# Liposomal Formulation of Botanical Extracts may Enhance Yield Triggering PR Genes and Phenylpropanoid Pathway in Barley (*Hordeum vulgare*)

**DOI:** 10.3390/plants11212969

**Published:** 2022-11-03

**Authors:** Géza Hegedűs, Barbara Kutasy, Márta Kiniczky, Kincső Decsi, Ákos Juhász, Ágnes Nagy, József Péter Pallos, Eszter Virág

**Affiliations:** 1Research Institute for Medicinal Plants and Herbs Ltd., Luppaszigeti Str. 4, 2011 Budakalász, Hungary; 2EduCoMat Ltd., Iskola Str. 12A, 8360 Keszthely, Hungary; 3Department of Information Technology and its Applications, Faculty of Information Technology, University of Pannonia, Gasparich Márk u. 18/A., 8900 Zalaegerszeg, Hungary; 4Institute of Metagenomics, University of Debrecen, Egyetem Square 1, 4032 Debrecen, Hungary; 5Department of Plant Physiology and Plant Ecology, Institute of Agronomy, Hungarian University of Agriculture and Life Sciences Georgikon, 7 Festetics Str., 8360 Keszthely, Hungary; 6Department of Microbiology and Applied Biotechnology, Institute of Genetics and Biotechnology, Hungarian University of Agriculture and Life Sciences, Páter Károly Str. 1, 2100 Gödöllő, Hungary; 7Department of Molecular Biotechnology and Microbiology, Institute of Biotechnology, Faculty of Science and Technology, University of Debrecen, Egyetem Square 1, 4032 Debrecen, Hungary

**Keywords:** ELICE16INDURES, biostimulant, gene expression, *Hordeum vulgare*, plant conditioner, Illumina RNA-seq

## Abstract

This work aimed to study the plant conditioning effect and mode of action of a plant-based biostimulant used in organic farming. This new generation plant biostimulant, named ELICE16INDURES^®^, is rich in plant bio-active ingredients containing eleven supercritical botanical extracts encapsulated in nano-scale liposomes. The dose–response (10 to 240 g ha^−1^) relationship was tested in a field population of autumn barley (*Hordeum vulgare)* test crop, and underlying molecular mechanisms were studied. Applying nanotechnology, cell-identical nanoparticles may help the better uptake and delivery of active ingredients increasing resilience, vitality, and crop yield. The amount of harvested crops showed a significant increase of 27.5% and 39.9% interconnected to higher normalized difference vegetation index (NDVI) of 20% and 25% after the treatment of low and high dosages (20 and 240 g ha^−1^), respectively. Illumina NextSeq 550 sequencing, gene expression profiling, and KEGG-pathway analysis of outstanding dosages indicated the upregulation of pathogenesis-related (PR) and other genes—associated with induced resistance—which showed dose dependency as well.

## 1. Introduction

The use of non-chemical biostimulants is the ecologically safest solution for increasing crop vitality and production. Today, sustainable agricultural production and the preservation of biodiversity in agricultural areas receive a lot of attention worldwide; therefore, the development of alternative crop protection aims to avoid the use of chemicals such as synthetic fungicides, insecticides, or herbicides [1]. The “metabolic enhancers” are biologically active, plant signaling molecules that are in focus on several company products [2,3]. These products are called biostimulants, whose main active agents are vitamins or phytochemicals such as phytohormones, amino acids, and/or their derivatives [4].

These agents may help in agricultural important cultures to enhance yield through the reinforcement of biochemical responses against biotic and abiotic stresses. The molecular effects of biostimulants were first studied in *Arabidopsis thaliana* (*A. thaliana*, mouse-ear cress). Toscano et al. and Xu et al. reported on 127 genes with significantly higher activity in plants treated by several marketed biostimulant products [5,6]. Selivanova et al. (2015) reported an increased yield in cucumber plants after the treatment of some biostimulant products due to the metabolic enhancement of plants [7]. Klokić et al. (2020) investigated the effect of biostimulants on tomato yield and found that 10 different biostimulants affected the total phenol and flavonoid contents of fruit, leading to higher antioxidant capacity [8]. Certain biostimulants are protective against a broad range of plant pathogens by activating plant immunity [9,10]. Usually, their effectiveness is due to potentiate plant defense mechanisms instead of direct pathogen targets. These reinforced defense responses may become permanent, called priming or the primed state. Substances that may trigger the priming in plants may be called imprimatin compounds [11]. From an economic aspect, this kind of induced resistance should be one of the optimal strategies for the development of biostimulants. The impact of priming active elicitors on defense signaling has been proven in crops; therefore, their potential application in field practice scientifically supported the opening of new possibilities in sustainable plant protection [12].

In the Research Institute for Medicinal Plants and Herbs Ltd. Budakalász, Hungary, a plant-extract-based biostimulant product, ELICE16INDURES^®^, was developed and licensed containing high amounts of allelochemicals such as flavonoids, steroids, terpenoids, saponins, alkaloids and phenolic compounds which can stimulate, e.g., plant development, nutrient assimilation, photosynthesis or pathogen resistance. In this study, we report on this agent as a potential new defense-priming material. Its technological development aimed to reach a 150–200 nm size multilamellar liposome formulation of eleven CO2-extracted medicinal plants as a novelty in this field.

To reach the best absorption of bioactive compounds, a state-of-the-art process is used, namely, nanoparticles in the range of 100–500 nm in size [13,14]. Abdel-Aziz et al. and Dutta report on the positive impact of nanomaterials in a foliar application and seed priming [15,16]. Despite nano-liposome technology being a less studied area in botany, several studies investigated the use of liposomes. Karny et al. reported on the application of liposomes to supplement plant growth and overcome acute nutrient deficiency [17]. The main advantage of liposome formulation is the use of less active compounds to reach maximal bioavailability. In this study, we report firstly on a plant conditioner with nano-size liposome formulation that showed priming activity tested in the field and laboratory as well.

The presence of priming was described in different types of induced resistance suggesting its crucial role in many defense mechanisms [18]. The most widely studied mechanisms are systemic acquired resistance (SAR) and induced systemic resistance (ISR) which are phenotypically similar but show alterations in hormonal control and associated gene expression [19]. Among others, the salicylic acid-dependent SAR displays a high accumulation of pathogenesis-related (PR) gene products, whereas ISR affects genes in ethylene and jasmonic acid responses. Natural inducers may stimulate induced resistance leading to a stronger phenotypic appearance of plants however in most cases the underlying accurate genetic mechanisms remain in question and may overlap. It is known that plant defense response begins with stress-related plant hormones, e.g., salicylic acid, jasmonic acid, and ethylene [20], that may be used also as inducers [21]. Hormonal inducers depending on the concentration may have a dual role in the activation of plant defense responses, e.g., relatively low doses of exogenous salicylic acid may induce a direct response of defense genes that behave classical SAR genes and parsley defense genes; higher doses may induce another set of defense genes [22]. Hormonal inducers are also associated with the induction of general phenylpropanoid pathway-responsive genes to reinforce lignin biosynthesis against pathogen attack [23]. In this study, we focused on the monitoring of genes taking part in different transcriptomic events during priming mechanisms that may lie behind the resistance development in plants characteristic to ELICE16INDURES.

PR genes are raised by diverse abiotic and biotic stresses in primed plants [18,24]. Inducible defense-related PR proteins were first identified in tobacco infected with the Tobacco mosaic virus [25], and were later described in various plant species previously infected by viruses, bacteria, and fungi [26]. PR proteins have been classified into 17 families based on biological activity, localization, molecular weight, and isoelectric point [26,27,28]. Among them, the PR1 family is used as a marker of enhanced plant disease resistance; although its biochemical activity and mode of action remain unknown, it is the only PR that does not contain functional annotations [29]. The properties of the other PR families are well known, including β-1,3-glucanases (PR2), chitinases (PR3-4), thaumatin-like proteins (PR5), peroxidases (PR9), ribosome-inactivating proteins (PR10), thionins (PR13), nonspecific lipid transfer proteins (PR14), oxalate oxidase (PR15), and oxalate-oxidase-like proteins (PR16) [26,30]. The resistance mechanism against various economically important fungal pathogens was investigated in barley plants. The expression patterns of PR genes were examined in the early stages of infection in susceptible and resistant genotypes using the quantitative PCR method. The data showed that in the resistant genotypes, PR transcripts accumulated higher and earlier than in the susceptible genotypes; the most prominent differences were observed in PR1 and PR5 genes, which were strongly activated [31]. Recombinant chitinase protein (PR3-4) from seeds of *H. vulgare* significantly inhibited the growth of the hyphae of three phytopathogenic fungi (*Alternaria alternate, Rhizoctonia solani* and *Fusarium oxysporum*) in in vitro qualitative antifungal assays [32]. Furthermore, an increase in the abundance of peroxidases and PR-3 and PR-5 was observed after infection of barley ears with macroconidia suspension of *Fusarium graminearum* [33]. The activity of chitinase (PR3-4), β-1,3-glucanase (PR2), and peroxidases significantly raised in barley after *Drechslera teres* (*Pyrenophora teres*) infection [34] and *Bipolaris sorokiniana* infection [35]. The effect of cadmium on protein expression was investigated by proteomic analysis in leaf apoplast proteins of barley seedlings. PR1 proteins, certain 1-3-glucanases (PR2), chitinases (PR3), members of the chitin-binding PR4 family, Thaumatin-like proteins (PR5) and PR17 proteins were identified, indicating that this abiotic stress-activated a general plant defense response using PRs [36]. Peroxidases are attended in lignification, regulation of cell wall elongation, wound healing, and resistance to pathogens in plants [37]. Caruso et al. (2001) [38] reported that heme peroxidase purified from wheat kernels reduced the elongation of the germ tube of *Fusarium culmorum, Trichoderma viride* and *Botrytis cinerea*.

The phenylpropanoid biosynthetic pathway contributes to several aspects of plant responses toward biotic and abiotic stimuli that might be part of priming mechanisms [39]. The increased synthesis of polyphenols under abiotic stress conditions helps plants to cope with different environmental conditions [40]. Among the key phenolic biosynthesis enzymes are phenylalanine ammonia-lyase (PAL), cinnamyl-alcohol dehydrogenase (CAD), and 4-coumarate-CoA ligase (4CL). PAL catalyzes the first step in the biosynthesis of phenylpropanoids and the PAL-related gene (*Hv*PAL) was highly regulated among the genes showing higher expression under metal stress [41], under wounding stress [42], and under fungal infection [43] in barley. The CAD enzyme that can be involved in the maintenance of lignin synthesis increases in activity during several pathogen responses so that the lignin synthesis may participate in plant resistance against pathogens [44]. The induced expression of genes related to secondary metabolism (glutathione-S-transferase (GST), PAL and CAD) was demonstrated under salt and drought stress conditions in wild barley [45]. UV-B treatment and water deficit enhanced the activity of 4CL in barley leaves and this enhancement was positively correlated with the accumulation of flavonols and anthocyanins [46].

In the present study, our goals were: to (i) develop 100–150 nm MLV liposomes as a new formulation of 11 supercritical botanical extracts and their use as a biostimulant (named ELICE16INDURES); (ii) investigate the plant conditioning effect of this liposomal agent on the field population of autumn barley; (iii) for practical use, we aimed to determine dose–response by testing six concentrations and measuring yield parameters and photosynthetic activity; (iv) to understand underlying resistance mechanisms, genome-wide transcriptional profiling was in focus using the same barley test crops under controlled plant growth conditions; (v) in the phytotron experiment doses showing higher conditioning effects in the field were aimed to test and compare with each other to gain information on the comprehensive mechanism of action of this agent. Results obtained as these goals are considered new results for sustainable plant protection involving the field of new-generation biostimulant development.

## 2. Results

### 2.1. Liposome Formation of ELICE16INDURES in the Range of 100–200 nm

The 11 active agents of ELICE16INDURES were entrapped into small multilamellar vesicles (MLV) of 100–150 nm. These liposomes were prepared by using active trapping techniques. Transmission electron microscopy (TEM) was used to characterize the sizes and structures of liposomes in the range of 10–1000 nm. Figure 1 shows TEM records of liposomes of ELICE16INDURES plant conditioner.

### 2.2. Field Experiments

#### 2.2.1. Increase of Crop Yield in ELICE16INDURES Treated Plots

The amount of the harvested crop was corrected to 8% moisture content showing an increase following the ELICE16INDURES treatment. The yield increase was 27.5% and 39.9% after the dosage of (low) 20 g ha^−1^ and (high) 240 g ha^−1^, respectively. These values were significant (F-test, P10%). The measured yield values of four parcels and the variance table can be seen in Appendix A. Numerically, both the low and high investigated dosages positively affected the crop yield calculated to kg plots^−1^ and T ha^−1^ (Figure 2).

#### 2.2.2. Higher NDVI in ELICE16INDURES Treated Plots

The dose–response relationship was analyzed by determining the photosynthetic activity by calculating NDVI [47] from the near-infra-red drone records (Figure 3). NDVI gives an idea of the changes in vegetation activity and vegetation ratio of a crop population in the field. [48]. Applying NDVI the heterogeneity within the field can be assessed and draw attention to the occurrence of problems or changes such as wildlife damage, inland water, nutrient deficiency, prolonged germination, and plant disease [49]. The development of vegetation as a consequence of different varieties, hybrids, or different crop production technologies can be followed and compared by monitoring NDVI alterations [50]. Admittedly, these are not exact values, but they can give a clue to the condition of the plants. In this study, we used NDVI spectral analysis to gain information on the plant conditioning effect of ELICE16INDURES like photosynthetic properties as indicators of the physiological state of the investigated barley population [51]. The higher NDVI value suggested a larger area with enhanced photosynthetic activity in leaves that were detected at parcels 4 and 8 (Figure 3b). This state suggested healthier leaf surfaces or better assimilation of nutrients, water, or light. We hypothesized that these plants showing higher NDVI values have overcome pathogen-induced tissue damage.

### 2.3. RNA-seq Analysis of ELICE16INDURES Treated Plants in Phytotron

The optimal effect of biostimulant material is highly dependent on the dosage response of plants. Therefore, 16-day-old barely plants were sprayed with low and high dosages (20 and 240 g ha^−1^) of ELICE16INDURES to gain information about the dose dependence. The mode of action was investigated by applying Illumina NextSeq550 RNA-Seq after two days of treatments. NGS libraries were prepared of control, low- and high-dosage-treated plants. Combined assembly of these libraries was performed de novo and pairwise differential expression was studied with KEGG pathway analysis.

#### 2.3.1. De Novo Sequencing and Transcriptome Assembly

De novo assembly of the 3 × 30 M combined read sets (cleaned reads) has resulted in 16, 492 total transcripts and 13, 513 total genes. The percentage of GC was 51.71. The resulting reference transcriptome was analyzed at the gene level and deposited into the Mendeley database. Statistics of transcripts are detailed in Table 1. Transcript abundancies were analyzed creating CountTable where the total mapped reads were presented for each transcript gene. The CountTable was deposited in the Mendeley database. The distribution of counts of transcripts is presented in Figure 4a. Based on the CountTable data shared (present in all samples), and individual transcripts were determined in the three samples. Numerical data are visualized with a Venn diagram, as shown in Figure 4b.

#### 2.3.2. Functional Analysis and Classification of Unigenes

The transcripts at the gene level (unigenes) derived from the de novo assembled transcriptome assembly was subjected to additional validation and annotation. BLASTx search of homology against the National Center for Biotechnology Information (NCBI) non-redundant (nr) *Viridiplantae* database was functionally annotated resulting in 99.3% of aligned sequences. Blast hits (13,505), Gene Ontology (GO) mapping (11,263), and GO slim are indicated in Figure 5. GO slims were also determined of the full GO having a broad overview of the ontology content without the detail of the specific fine details.

#### 2.3.3. Pairwise Differential Expression

Pairwise differential expression analysis was performed using the CountTable data. Two experimental set up were analyzed: differentially expressed genes (DEGs) between (i) low treatment vs. control and (ii) high treatment vs. control with the following outputs: The number of total features was 13,513. (i) DEGs were (Probability > 0.9) 158, up-regulated (M > 0): 156 and down-regulated (M < 0): 12. (ii) DEGs were (Probability > 0.9): 92, up-regulated (M > 0): 86, down-regulated (M < 0): 6. Principal Component Analysis (PCA) of DEGs in the three samples was performed and plotted in Figure 6. To compare the distribution of differentially enriched GO-terms across the three investigated samples of *H. vulgare*, GO-term enrichment analysis using Fisher’s exact test was performed. Enriched bar charts of GO names of upregulated genes are detailed in Figure 7. According to these results contigs with GO names of response to stress, response to biotic stimuli, and response to stimulus were further analyzed and discussed. Common and individual contigs expressed in the three samples were filtered and grouped. These groups and information of GO names, contig numbers, enzyme names, and functions are detailed in Table 2.

#### 2.3.4. Pathway Analysis

Pathway of DEGs of pairwise analysis was performed using Kyoto Encyclopedia of Genes and Genomes (KEGG) database. The summary of pathways is detailed in Table 3. The previous data suggested a high pathogen and biotic stress response as a result of high and low dosage of ELICE16INDURES; therefore, the phenylpropanoid biosynthetic pathway-containing the largest number of DEGs among the sequences aligned to the pathway-are designed and reported in this study (Figure 8). The 73 contigs were aligned to the phenylpropanoid pathway among which 9 showed differentiated expression, (8 up and 1 down-regulation). According to functional identification of these contigs oxidoreductases, CAD, PAL, 4CL, long-chain-fatty-acid-CoA ligase (LACS), beta-glucosidase, and glucan endo-1,3-beta-D-glucosidase were functionally identified. Phenylpropanoid biosynthesis KEGG pathway showed the most characteristic to both treatments. Therefore, genes of this pathway were selected for further analysis.

## 3. Discussion

In recent decades, the innovation of environmentally friendly natural plant biostimulants has been proposed to enhance the sustainability of agricultural production systems [52]. The development of products with biologically active agents focuses on improving yield and abiotic stress tolerance therefore, in this study, the effects of the newly developed liposomal formulation of SC-CO_2_ botanical extracts (ELICE16INDURES) were examined. This formulation technology was successfully applied in our previous research, suggesting the easier uptake of the active ingredients [53].

In this work the beneficial effect of the newly developed liposome-formulated plant biostimulant product, ELICE16INDURES was investigated and approached from two aspects. Firstly, agricultural parameters involving photosynthetic activity and yield measurements were analyzed in field conditions by applying drone technology and standard agronomical measurements for yield calculation. Interestingly two extremal treatments (low and high, 20 and 240 g ha^−1^) showed the strongest effect. Secondly, using the low and high concentrations, the genetic background of the physiological state of plants was investigated by applying genome-wide transcriptional profiling. Based on the knowledge of genetic and phenological alterations, the possible priming effect of this agent was also hypothesized and analysed deeper.

Using NDVI spectral analysis, the photosynthetic traits and plant vitality may be detected and plant responses to biotic and abiotic stresses may be assumed by the phenological appearance. We found the highest NDVI values in the parcels treated by 20 and 240 g ha^−1^ dosages of ELICE16INDURES. This state suggested fewer patients and/or better assimilation of nutrients, water, or light. The higher NDVI values indicated healthier leaf surfaces detected in these parcels that were based on a better response to pathogens. This pathogen response was proved by the higher expression of PR genes in phytotron experiments. The positive physiological alteration of barley cultures was manifested also in practical aspects in the yield increase, where the dose-dependency was also experienced, and found that 20 and 240 g ha^−1^ to be the most effective for yield growth.

Barley plants treated with plant biostimulant ELICE16INDURES of different doses were shown overexpressed defense response-related genes detected by the whole genome transcriptome analysis. Using pairwise differential expression analysis between treated and untreated samples, overexpressed PR genes were detected, which usually activate biotic and abiotic stress-response reactions [30,54]. Among these PRs, the PR1 and PRB1-2 genes were shown an increased expression level, which encodes proteins whose exact mode of action is unknown, but has already been described as having antifungal properties [31] and as an adequate plant response to cadmium stress [36] in barley. The non-race-specific disease-resistance (NDR1/HIN1) protein was also activated in the samples, which was previously described in the *A. thaliana* plants after bacterial infection when it increased together with PR1 as a signal recognition of the plant response to pathogen infection [55]. Moreover, the PR2, PR3, PR4, and PR5 genes showed a significantly increased expression level that phenomena were detected in barley after various fungal infections [31,32,33,34] and abiotic stress [36]. The increased activity of different peroxidases in the plants can be involved in the resistance to pathogens and in antioxidation [34,37,38]. Certain peroxidases belong to PR9 family which were observed with significant changes in expression levels after *Fusarium graminearum* infection in barley [56]. Similarly, the raised level of peroxidases (peroxidas1, 2, 5, and 50) was detected in our samples. Transcription factors (TFs) were included among DEG Top50 genes, the member of TIFY and WRKY families. TIFYs play an essential role in cross-talk between jasmonic acid (JA) and phytohormones signaling pathways [57]. The specific TIFY11-like protein involved in the response to wounding and JA-mediated immune response [58] indicated a strong upregulation after two days of ELICE16INDURES treatment in field conditions. The WRKY proteins are involved in defense response to abiotic and biotic stresses modulating gene expression levels [59]. *Hv*WRKY24 belonging to group III. [60] was upregulated under salt stress [61], as was detected in barley samples after ELICE16INDURES treatments.

Transcripts that showed a significant difference in the DEG analysis were further analyzed with the KEGG database; as a result, the phenylpropanoid biosynthetic pathway and starch and the sucrose metabolism pathway displayed a higher activity after both treatments. Phenolic compounds belonging to secondary metabolites are produced under optimal and suboptimal conditions in plants. Phenolic biosynthesis can participate in stress response under abiotic stress conditions [40], as was observed in our samples. After ELICE16INDURES treatments, the activity of key enzymes induced, as PAL, CAD, and 4CL genes were upregulated. These phenomena are well-described in barely under stress conditions [41,42,43,45,46]. In higher plants, the major storage carbohydrate is starch which is essential for sustaining metabolism, growth, and development when photosynthesis is not active [62]. Among enzymes of starch metabolism, the starch phosphorylase (alpha-glucan phosphorylase) was shown a higher activation in barley samples that plays a key role in the mobilization of stored polysaccharides catalyzing polysaccharides into α- d -glucose-1-phosphate [63]. Moreover, overexpressed glucan endo-1,3-beta-glucosidase genes were observed after treatments, which have been implicated in diverse physiological and developmental processes and defense against biotic and abiotic stress [64,65,66]. All of the induced genes were described in connection with the defense-priming [24,67] indicating the priming property of ELICE16INDURES.

Transcriptomic analysis of the dose–response relationship showed different roles in induced resistance triggered by ELICE16INDURES low and high concentration, even though similar phenotypical appearances were observed in both treatments. Similar phenomena were described in plant hormonal treatments, salicylic acid, and other hormone analogs as dual roles in SAR inducers. These exogenous treatments activated plant defense responses and induced resistance however low and high doses influenced ISR (with genes involved in jasmonate and ethylene response) and SAR (with the accumulation of PRs), respectively [18,22,68,69]. Similar gene activation was observed in our results, suggesting that low dosage (20 g ha^−1^) may activate the elicitors of defensive genes (e.g., ethylene-responsive genes and leading to primed plants); however, high dosage (240 g ha^−1^) may directly induce PR genes. (This theory is summarized in Figure 9). Furthermore, the high concentration of ELICE16INDURES suggested a higher exogenous phytohormonal (derived from plant extracts) interaction with cells influencing intracellular hormonal amount. This may result in elicited defense responses, including the activation few PR genes observed in high-dosage samples. Similar phenomena were described reporting on a low and high concentration of salicylic acids affecting elicitors and SA-depending genes like PAL and 4CL [22]. The potential role of ELICE16INDURES dosages in ISR and SAR mechanisms is planned to be analyzed in time course experiments in controlled phytotron and in vitro conditions applying different phytopathogenic infections. According to this study, the priming activity of ELICE16INDURES has been proven, we also plan to investigate the plant taxonomical dependency of dose–response. Taxonomical dependency of a priming active material (β-aminobutyric acid) was reported by our previous work comparing autumn barley and *A. thailana* showing altered transcriptomic responses to the drug [70]. The results of this study may be involved in the practical use of ELICE16INDURES and guide the farmers to apply this product for symptomatic, preventive, or long-term treatment.

Transcriptomic analysis of dose–response showed different roles in induced resistance triggered by ELICE16INDURES low and high concentration, even though similar phenotypical appearances were observed in both treatments. Similar phenomena were described in plant hormonal treatments, salicylic acid, and other hormone analogs as a dual role in SAR inducers. These exogenous treatments activated plant defense responses and induced resistance however low and high doses influenced ISR (with genes involved in jasmonate and ethylene response) and SAR (with the accumulation of PRs), respectively [18,22,68,69]. Similar gene activation was observed in our results, suggesting that low dosage (20 g ha^−1^) may activate the elicitors of defensive genes (e.g., ethylene-responsive transcription factors and lead to primed plants) however high dosage (240 g ha^−1^) directly induce PR genes (This theory is summarized in Figure 9). Furthermore, the high concentration of ELICE16INDURES suggested a higher exogenous phytohormonal (found in plant extracts) interaction with cells influencing intracellular hormonal amount. This may result in elicited defense responses, including the activation few PR genes observed in high-dosage samples. Similar phenomena were described reporting on a low and high concentration of salicylic acids affecting elicitors and SA-dependent genes like PAL and 4CL [22]. The potential role of ELICE16INDURES dosages in ISR and SAR mechanisms is planned to analyze deeper in time course experiments in controlled phytotron and in vitro conditions applying different phytopathogenic infections. According to this study, the priming activity of ELICE16INDURES has been proven, we also plan to investigate the plant taxonomical dependency of doses. Taxonomical dependency of a priming active material (β-aminobutyric acid) was reported by our previous work comparing autumn barley and *A. thailana* showing altered transcriptomic responses to the drug [70]. The results of this study may be involved in the practical use of ELICE16INDURES and guide the farmers to apply this product for symptomatic, preventive and/or long-term treatment.

To summarize our results and findings, (i) a 100–150 nm size MLV liposome-formulated botanical extracts-containing biostimulant, ELICE16INDURES, was developed and used in field practice; (ii) the plant conditioning effect of this agent was proved in four repetitions block system of autumn barley test crop; (iii) for practical use, dose–response of plants was determined showing two effective (extremal) dosages, 20 and 240 g ha^−1^, where the yield and the photosynthetic activity increased by 39.9% and 27.5%; (iv) we proved that these concentrations stimulate plant defense responses increasing the expressions of PR and phytohormone-signalling pathway genes and transcription factors; (v) comparing the expressed genes and transcriptomic events between the two extremal dosages, we found that this biostimulant possesses priming effect on the investigated test crop and may enhance ISR and SAR mechanisms depending on low and high concentrations.

## 4. Materials and Methods

### 4.1. Preparation of ELICE16INDURES Plant Conditioner

High-pressure extracts with supercritical carbon dioxide, SC-CO_2_ extraction as a solvent of eleven botanical extracts were purchased by FLAVEX Naturextrakte GmbH, Germany. Extracts used to prepare the product ELICE16INDURES are detailed in Appendix A. Extracts of medicinal plants were established in a common set of states and encapsulated in plant-lecithin-based liposomes. These active agents were entrapped into MLV of 150–200 nm by using active trapping techniques [71]. The nanoparticle size distribution was measured by TEM recording.

### 4.2. Cultivation of Plants and Treatment

Fresh leaves were collected on day two after treatments from 16-day-old autumn barley, variety ‘SU Ellen’, plants (diploid). Plants were cultivated in phytotron and arable fields as detailed below.

The field experiment was performed during the vegetation period of 2020–2021 in the experimental field of Plant-Art Research Ltd., Hungary, where the autumn barley variety ‘SU Ellen’ was used as a testing crop. Plants in field experiments were sprayed by TTAM4E drone using a range of low to high (10 to 240 g ha^−1^) doses of ELICE16INDURES. We used a commercially available positive control plant conditioner, Fitokondi^®^ (4 L ha^−1^). Applied doses and plot allocations were: 1, control without treatment; 2, positive control treated with Fitokondi; 3, 10 g ha^−1^; 4, 20 g ha^−1^; 5, 30 g ha^−1^; 6, 60 g ha^−1^; 7, 120 g ha^−1^; 8, 240 g ha^−1^ ELICE16INDURES. The placement of parcels is shown in Figure 3. Test culture and site characteristics are presented in Appendix A. Description of ELICE16INDURES application conditions and technique are detailed in Appendix A. Yield measurements per plot were performed by instrumental measurement of hectolitre weight using XGrain (Infracont, Pomáz, Hungary). The experimental design of the field experiment is summarized in Figure 10a.

Exogenous treatment of ELICE16INDURES was performed in phytotron experiments. Plant growth chamber type was MLR352HPA −115V NEMA 5–20, 220 V/60 Hz–Panasonic. Treatment conditions were as follows: temperature during the first day and night was 25 °C. The temperature during the 2–16 days and nights was 25 °C and 15 °C, respectively. Duration of the day was 12 h, 04–4 p.m. Treatments were as follows: low dosage, 0.1 mL/1000 mL water, and high dosage, 1 mL/1000 mL water, corresponding to 20 and 240 g ha^−1^ spraying at the field. The experimental design of the phytotron experiment is summarized in Figure 10b.

### 4.3. Determination of NDVI by Remote Sensing

To monitor NDVI in the field population we used a DJI-phantom 4 Pro agro drone equipped with a near-infrared camera. The recording was carried out on the twelfth day after the first treatment (17 May 2021). Single aerial pictures were combined afterward with AgiSoft Photoscan Professional software using high-quality dense cloud processing and mesh construction settings (Figure 3). For NDVI calculation after identifying the plots based on combined aerial photographs, sample areas were cut out using self-developed software. Sample collection for NGS gene expression profiling was performed on the NDVI recording day.

### 4.4. RNA Extraction

Approximately 30 mg of plant tissue was placed in a 1.5 mL Eppendorf LoBind tube containing glass beads 1.7–2.1 mm diameter (Carl Roth, Karlsruhe, Germany) and 100 µL of TRI-Reagent (Zymo Research, Irvine, US). The Eppendorf tube was firmly attached to a SILAMAT S5 vibrator (Ivoclar Vivadent, Schaan, Liechtenstein) to disrupt and homogenize the tissue for 2 × 15 s. Total RNA was extracted using Direct-zol™ RNA MiniPrep System (Zymo Research, Irvine, CA, USA) according to the manufacturer’s protocol. The RNA Integrity Numbers and RNA concentration were determined by RNA ScreenTape system with 2200 Tapestation (Agilent Technologies, Santa Clara, CA, USA) and RNA HS Assay Kit with Qubit 3.0 Fluorometer (Thermo Fisher Scientific, Waltham, MA, USA), respectively.

### 4.5. Preparation of RNA-seq Libraries

For genome-wide gene expression profiling, three RNA-seq libraries such as control plants and treated with the low and high dosages of ELICE16INDURES were prepared from phytotron cultivated samples. Pooled samples were taken from four individuals. For poly-A based mRNA enrichment and cDNA synthesis, the Illumina TruSeq™ RNA sample preparation kit (Low-Throughput protocol) was used according to the manufacturer’s instructions. The RNA sequencing was performed using Illumina NextSeq550 system. The samples were run using multiple indexing adapters. For library amplification, an adapter-selective PCR reaction was performed. The size and purity of the samples were checked by Agilent 2100 Bioanalyzer (Agilent Technologies, Santa Clara, CA, USA). The DNA libraries were multiplexed, normalizing them to 10 nM.

### 4.6. Bioinformatics Analysis–Read Processing

Libraries were sequenced with a final output single-end, 30 M × 80 bases long. Quality control (QC), trimming, and filtering of fastq files was performed in preprocessing step. The QC analysis was performed with FastQC software [72]. The Phred-like quality scores (Q scores) were set to >30. Poor quality reads, adapters at the ends of reads, and limited skewing at the ends of reads were eliminated by using Trimmomatic [73]. Contamination sequences and Ns were filtered out with a self-developed application GenoUtils as described earlier [74] reads containing Ns more than 30% were eliminated; reads with lower N’ ratio were trimmed with a final length > 65. Reads passed of preprocessing were further assembled and analyzed.

### 4.7. De Novo Assembly of Combined Read Sets

Reference transcript datasets were first created from the phytotron and field libraries using Trinity assembler with 23K-mer [75]. For de novo assembly and mapping, we used a server with 512 GB (Gigabytes) of RAM, 64 cores (CPUs), and Ubuntu as the operating system. To assess the read composition of the assembly, input RNA-Seq reads were aligned to the transcriptome assembly using Bowtie2 [76]. Reads mapped to the assembled transcript were captured. For gene expression profiling collapsing of splicing isoforms was performed with omixbox.biobam (OmicsBox, https://www.biobam.com/omicsbox, accessed on 1 September 2021) and SuperTranscripts (gene-level assembly) were used for gene expression investigations. Reference transcript datasets were deposited in the Mendeley database under the accession DOI:10.17632/mfg64trszy.1.

### 4.8. Functional Annotation

AnnotationTable including functional annotation of the entire de novo transcriptome (based on gene level) was performed with GO analyses using OmicsBox.BioBam (v2.0) as detailed by Decsi et al. 2022 [77]. In this step, the Blastx-fast with a permissive expectation value of 1 was used. GO IDs, GO names, GO slim, Enzyme codes, and Enzyme names were determined and written into the AnnotationTable. AnnotationTable was deposited in Mendeley data under the accession DOI:10.17632/mfg64trszy.1.

### 4.9. Gene Level Quantification

To estimate gene expression from RNA-sequencing, CountTable was created. To count how many reads map to each feature of interest (genes) each sample reads were aligned to the reference de novo assembled transcripts at the gene level. CountTable creation was performed with omixbox.biobam using the HTseq package [78]. Based on the data from CountTable, further analyses were performed, such as differential expression and gene set enrichment analysis. CountTable was deposited in Mendeley data under the accession DOI:10.17632/mfg64trszy.1.

### 4.10. Pairwise Differential Expression Analysis and KEGG Pathway Analysis

Numerical analysis of DEGs in a pairwise comparison of two different experimental conditions—gene expression analysis—was carried out using omixbox.biobam. The used application is based on the NoiSeq program that implements quantitative statistical methods to evaluate the significance of individual genes between two experimental conditions [79]. RPKM (Reads Per Kilobase per Million mapped reads) normalization method was performed.

### 4.11. Enrichment Analysis

The gene set enrichment analysis was performed according to the GSEA computational method defining sets of genes as statistically significant and showing differences between two biological states consistently [80]. The GSEATable was performed by using OmicsBox.BioBam (v2.0) and deposited in Mendeley data under the accession DOI:10.17632/mfg64trszy.1

### 4.12. Accession Numbers

Raw reads of this project used for phytotron and field experiments are deposited in the NCBI SRA database under the accessions: Bioproject, PRJNA721578, https://www.ncbi.nlm.nih.gov/bioproject/PRJNA721578; accessed on 1 September 2022, RNA sequencing of phytotron experiments, SRX10603947-SRX10603949; Sequencing of field experiments, SRX10598683, SRX10598684. Super transcripts. fasta, CountTable, AnnotationTable, and GSEATable are deposited to the Mendeley Data under the Doi: 10.17632/mfg64trszy.1, the direct link to these datasets: https://data.mendeley.com/datasets/mfg64trszy accessed on 1 October 2022.

## 5. Conclusions

In conclusion, the application of different dosages of the plant biostimulant ELICE16INDURES was tested in the field population of autumn barley. In this study, we report firstly on the practical use of 100–150 nm liposome-formulated plant-extract-based biostimulant material. The plant conditioning effect of this agent was observed at relatively low and high dosages (20 and 240 g ha^−1^) that were phenotypically manifested by increased yield and NDVI. Based on field experiments, the two outstanding dosages were selected for RNA-Seq experiments that were performed in a controlled condition of phytotron. The pairwise transcriptional profiling indicated many pathogenesis-, resistance- and defense-priming-related genes involved in the top 50 expressed. Based on these genes, the dose–response analysis suggested that the investigated conditioner concentrations have a different role in induced resistance affecting ISR, SAR, and priming mechanisms. KEGG pathway analysis highlighted that the most affected pathways (DEGs) were the phenylpropanoid and starch and sucrose pathways. Most of their genes may be associated with defense pathways, which was proved by pairwise expression analysis. Results reinforced the hypothetical priming activity of ELICE16INDURES; however, a comprehensive analysis of pathogenic infection should be performed to prove this. Additionally, a taxonomical dependency of selected concentrations is hypothesized therefore further investigations are planned to supplement the results of this study. The results on dose-dependency may help agronomists to optimal use of ELICE16INDURES. The obtained transcriptomic data contribute to the development of new-generation biostimulants used in organic farming.

## Figures and Tables

**Figure 1 plants-11-02969-f001:**
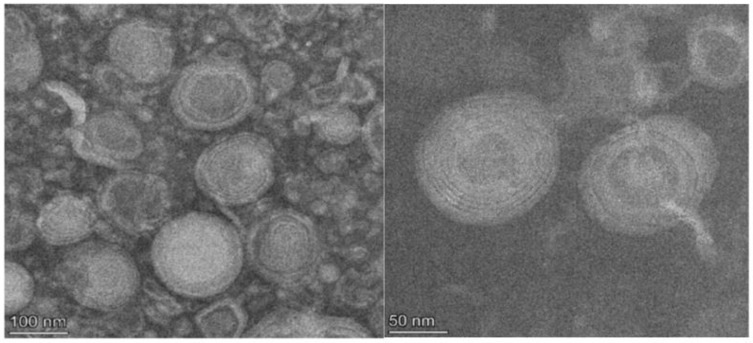
TEM records of the structure of ELICE16INDURES liposomes. TEM recordings showed a multilamellar liposome structure in the range of 100–200 nm.

**Figure 2 plants-11-02969-f002:**
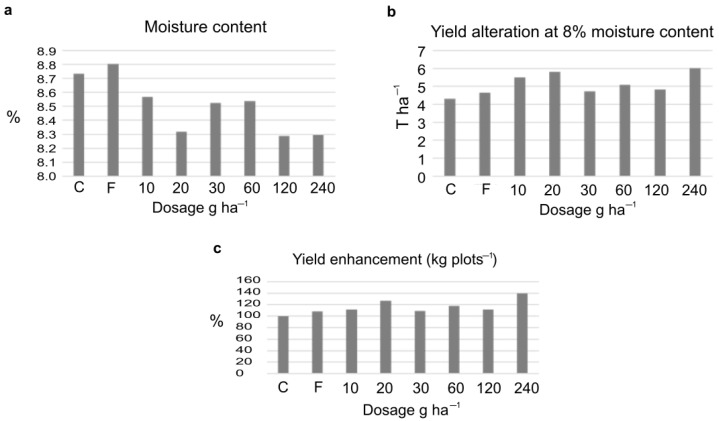
Changes of crop yield after treatments of 10–240 g ha^−1^dosage of ELICE16INDURES. The commercially available plant conditioner, Fitokondi (F) was used as a positive control (4 L ha^−1^). (**a**) Determination of moisture calculating mean values of four parallel parcels (%), moisture values were not significant according to the F-probe. (**b**) Determination of crop yield at 8% moisture content, calculating the mean values of four parallel parcels (T ha^−1^). According to the F-test, these values were significant at the P10%. (**c**) Determination of mean values of crop measured in four parallel parcels in kg plots ^−1^; to better understand, the yield enhancement is represented in percent increase in the diagram. According to the F-test, these values were significant at the P10%. A significant increase in yield was detected in the two investigated dosages of 20 and 240 g ha^−1^.

**Figure 3 plants-11-02969-f003:**
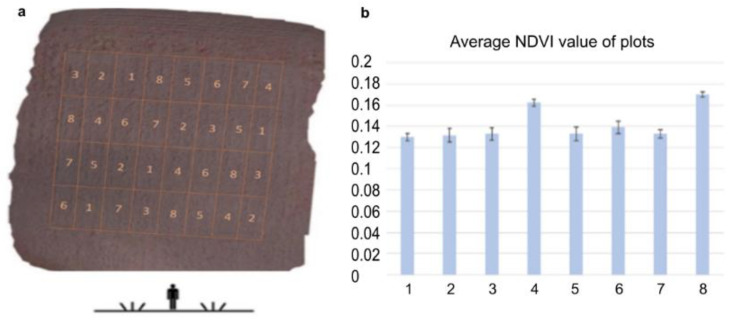
The agro-drone record was captured with a near-infrared camera during the experimental parcels. (**a**) Placement of parcels are: (1) control without treatment; (2) positive control treated with Fitokondi; (3) 10 g ha^−1^; (4) 20 g ha^−1^; (5) 30 g ha^−1^; (6) 60 g ha^−1^; (7) 120 g ha^−1^; (8) 240 g ha^−1^ ELICE16INDURES (GPS coordinates: 47 deg37′58.30″ N, 18 deg15′54.36″ E). (**b**) Average NDVI values of plots calculated from near-infrared channels of Agro-drone. The highest NDVI values were shown after the low and high (20 and 240 g ha^−1^) dosages of ELICE16INDURES treatment.

**Figure 4 plants-11-02969-f004:**
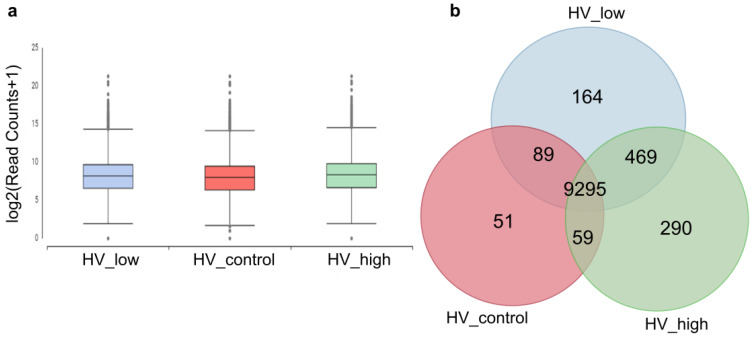
Numerical data of transcripts and their abundancies between the three samples. (**a**) Distribution of read counts of transcripts at the gene level. (**b**) Venn diagram: number of transcripts present in the investigated samples and unigenes. Marks: control (red: HV_control), treated samples with a high and low dosage of ELICE16INDURES (green: HV_high and blue: HV_low).

**Figure 5 plants-11-02969-f005:**
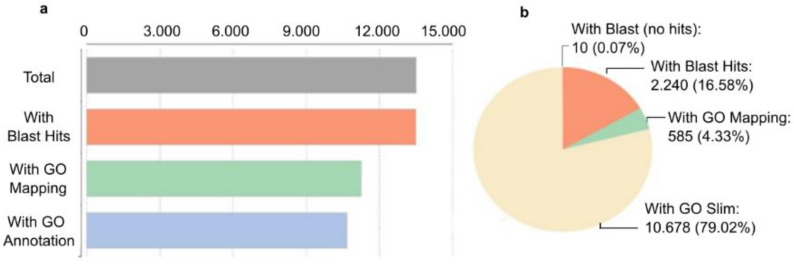
Characterization of *H. vulgare* de novo reference transcriptome. Transcripts at the gene level were annotated based on NCBI non-redundant (nr) *Viridiplantae* database search (downloaded in 2022.03). To annotate genes with GO slim terms, annotations and search all ancestors of the terms up to the root of the ontology tree were determined. The ancestor’s terms which are part of the slim subset were selected. (**a**) The number of annotated transcripts. (**b**) Percent distribution of annotation hits.

**Figure 6 plants-11-02969-f006:**
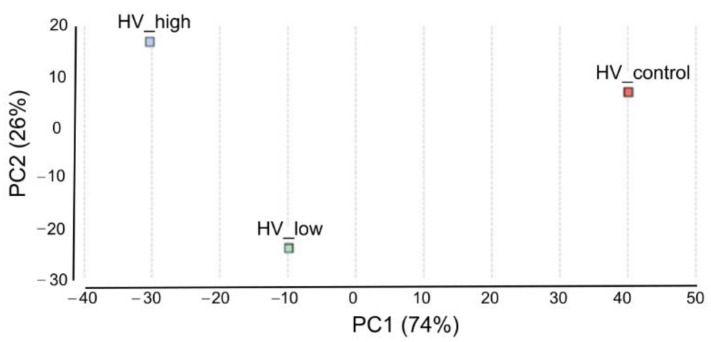
Principal Component Analysis of DEGs. Scatterplot of PC1 and PC2, explaining 74% and 26% of the variation, respectively, and separating the samples according to ELICE16INDURES treatment. The color of the data points indicates the treatments (red: control, blue: high dosage, and green: low dosage).

**Figure 7 plants-11-02969-f007:**
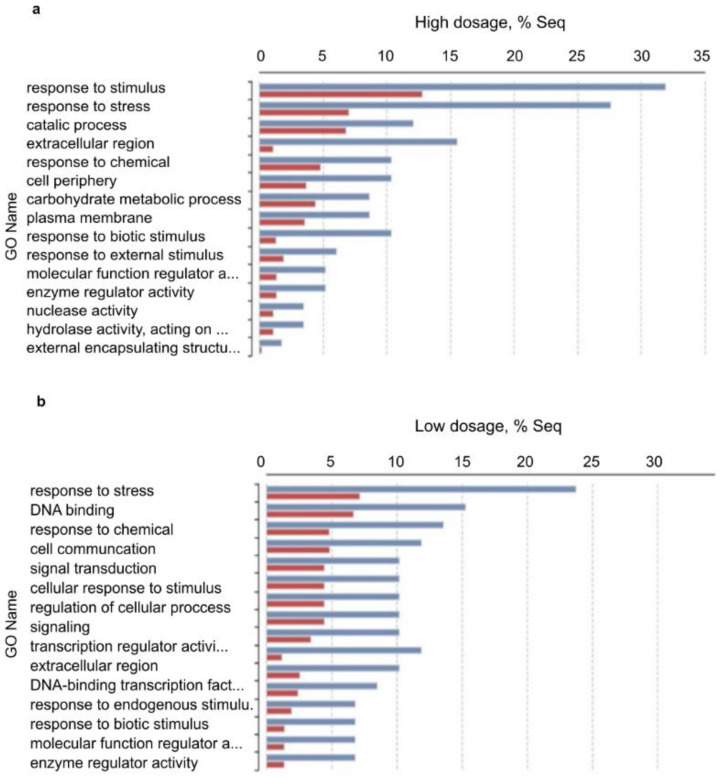
Enriched bar chart of GO names of up-regulated genes as results of pairwise DEGs analysis (Fisher’s exact test). Symbols: (**a**) upregulated GOs of high-dosage treated plants vs. control; (**b**) upregulated GOs of low-dosage treated plants vs. control. Red: treated samples; Blue: control samples. According to these results, contigs of response to stress, response to biotic stimuli, and response to stimulus GO names were further analyzed.

**Figure 8 plants-11-02969-f008:**
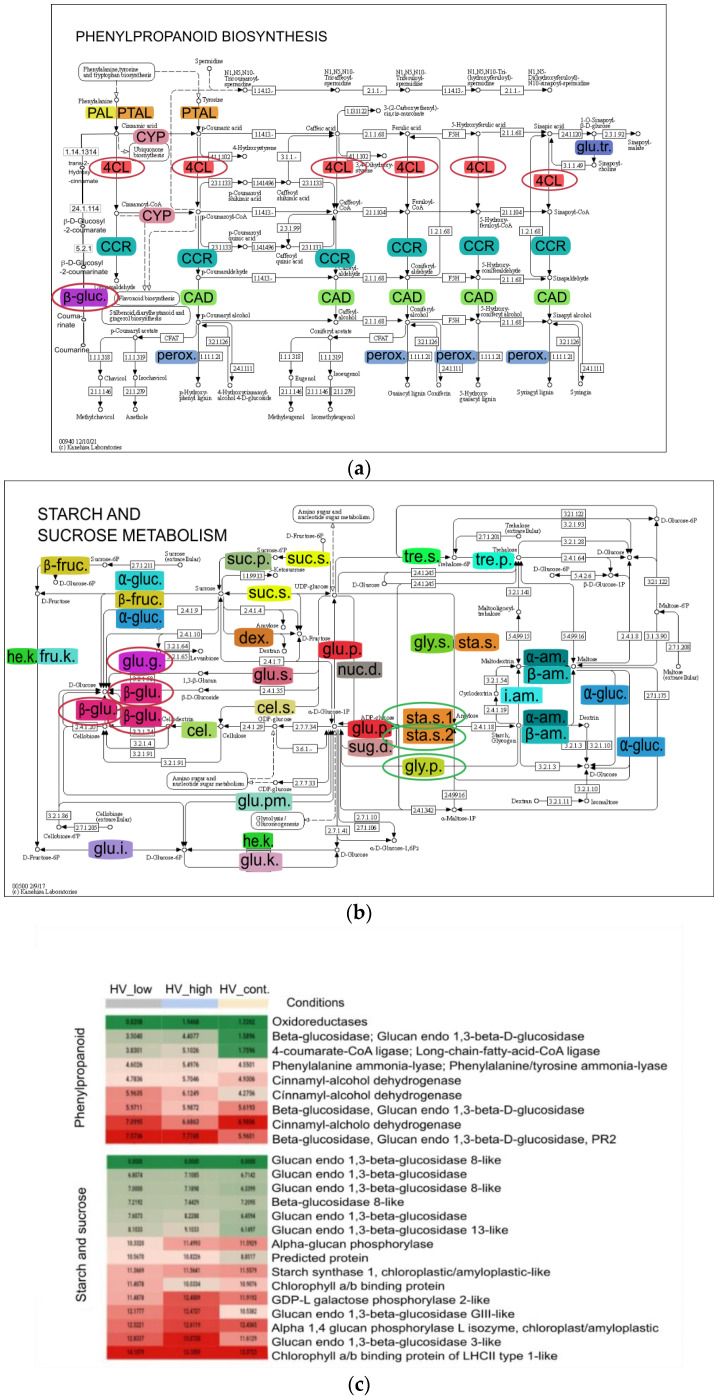
(**a**) Phenylpropanoid and (**b**) starch and sucrose biosynthetic pathways with DEGs. In this KEGG pathway analysis, 73 and 142 contigs were aligned; among them, 9 and 5 showed differentiated expressions that are functionally identified, respectively. Colored enzyme codes mean mapped contigs to the pathways that were found in samples treated by ELICE16INDURES (low and high dosage). (**c**) Heat map highlights the up- and downregulated genes found in these pathways circled with red and green in KEGG-maps. Abbreviations of phenylpropanoid pathway: phenylalanine ammonia-lyase (PAL), phenylalanine/tyrosine ammonia-lyase (PTAL), trans-cinnamate 4-monooxygenase (CYP), 4-coumarate--CoA ligase (4CL), cinnamoyl-CoA reductase (CCR), cinnamyl-alcohol dehydrogenase (CAD), beta-glucosidase (β-gluc.), sinapate 1-glucosyltransferase (glu.tr.) and peroxidase (perox.) Abbreviations of starch and sucrose metabolism: beta-fructofuranosidase (β -fruc.), alpha-glucosidase (α-glu.), sucrose alpha-glucosidase, hexokinase (he.k.), fructokinase (fru.k.), glucokinase (glu.k.), glucose-6-phosphate isomerase (glu.i.), glucose phosphomutase (glu.pm.), glycogen phosphorylase (gly.p.), ADP-sugar diphosphatase (sug.d.), glucose pyrophosphorylase (glu.p.), starch synthase (sta.s.), cellulose synthase (cel.s.), 3 beta-glucan synthase (glu.s.), cellulose (cel.), beta-glucosidase (β-glu.), glucan endo-1,3-beta-D-glucosidase (glu.g.), dextransucrase (dex.), sucrose synthase (suc.s.), sucrose-phosphate phosphatase (suc.p.), trehalose phosphate synthase (tre.s.), trehalose-phosphatase (tre.p.), nucleotide diphosphatase (nuc.d.), UDP-glycogen synthase (gly.s.), alpha-amylase (α-am.), beta-amylase (β-am.) and isoamylase (i.am.).

**Figure 9 plants-11-02969-f009:**
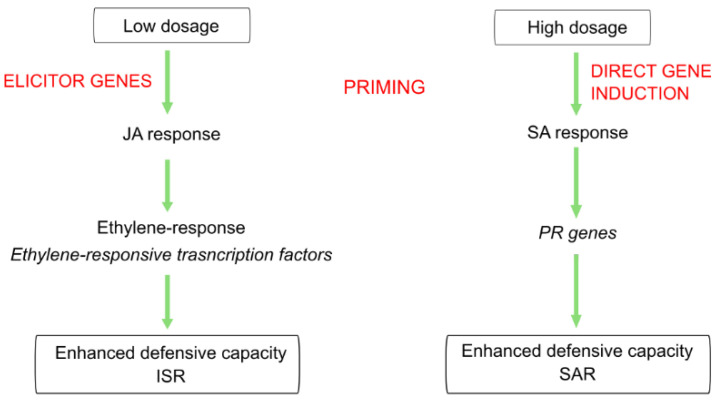
Summary of the hypothesized role of different dosages of ELICE16INDURES during two induced resistance mechanisms.

**Figure 10 plants-11-02969-f010:**
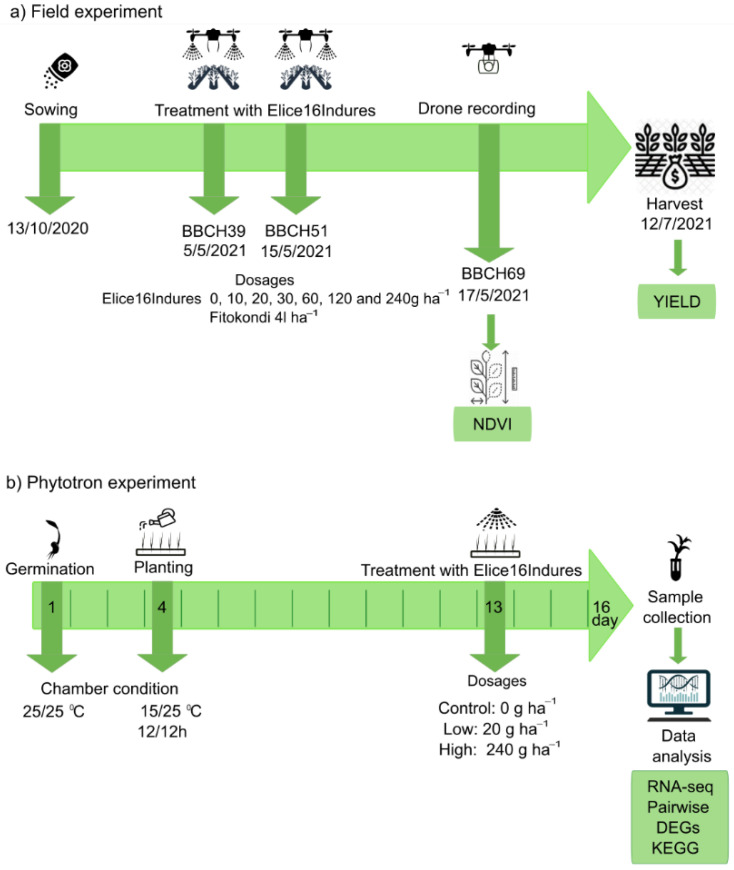
Design of field and phytotron experiments. Green rows indicate the timeline of plant cultivation, plant phenophases, treatments, harvest, and sample collection of autumn barley in the field (**a**) and phytotron (**b**) experiments. Green boxes indicate which measurements were performed in the experiments.

**Table 1 plants-11-02969-t001:** Contig length distribution of reference transcripts assembled of combined read sets of control, high and low dosage treated samples in the phytotron.

Contig Length	Stats Based on All Transcripts	Stats Based on the Longest Isoform Per Gene
N10	3271	3204
N20	2501	2466
N30	2074	2054
N40	1783	1765
N50	1538	1524
Median	1023	997
Average	1232.5	1209.74

**Table 2 plants-11-02969-t002:** Up-regulated genes of GO categories *Response to stress, Response to biotic stimuli,* and *Response to stimulus* based on Fisher’s exact test. Shared (present in both treated samples) and individual up-regulated genes of high and low dosage treatment of ELICE16INDURES are presented.

Contig ID	Enzyme Description	GO Names
Shared		
TRINITY_DN10333_c0_g1 TRINITY_DN370_c0_g1	NDR1/HIN1-like protein 10	P:response to stress; P:response to external stimulus; P:response to biotic stimulus; C:plasma membrane
TRINITY_DN1083_c0_g1 TRINITY_DN23_c1_g1	peroxidase 1-like root peroxidase	P:response to stress; P:catabolic process; P:cellular process; P:response to chemical; F:catalytic activity; F:binding; C:extracellular region
TRINITY_DN10948_c0_g1	putative protease inhibitor	P:response to stress; P:protein metabolic process; F:enzyme regulator activity
TRINITY_DN10986_c0_g1	asparagine synthetase (glutamine-hydrolyzing)	P:response to stress; P:cell communication; P:biosynthetic process; P:response to external stimulus; P:response to chemical; F:nucleotide binding; F:catalytic activity
TRINITY_DN2054_c0_g1	Subtilisin-chymotrypsin inhibitor-2B	P:response to stress; P:protein metabolic process; F:enzyme regulator activity
TRINITY_DN28_c0_g1 TRINITY_DN914_c0_g1	glucan endo-1,3-beta-glucosidase GIII-like (PR2)	P:carbohydrate metabolic process; P:response to stress; F:hydrolase activity; C:plasma membrane
TRINITY_DN3218_c0_g1	protein TIFY 11b-like (TF)	P:response to stress; P:signal transduction; P:response to endogenous stimulus; P:response to chemical; C:nucleus
TRINITY_DN326_c0_g1 TRINITY_DN5974_c0_g1	WRKY transcription factor (TF) WRKY transcription factor WRKY24-like	P:response to stress; P:signal transduction; P:response to external stimulus; P:response to biotic stimulus; P:response to abiotic stimulus; P:response to endogenous stimulus; P:response to chemical; F:DNA binding; F:chromatin binding; F:DNA-binding transcription factor activity; C:nucleus
TRINITY_DN505_c0_g1 TRINITY_DN505_c0_g2	pathogenesis-related protein 1 (PR1)	P:response to stress; P:response to biotic stimulus; C:extracellular region
TRINITY_DN505_c0_g3	pathogenesis-related protein PRB1-2-like	P:response to stress; P:response to biotic stimulus; C:extracellular region
TRINITY_DN7081_c0_g1 TRINITY_DN7081_c0_g2	nematode resistance protein-like HSPRO1	P:response to stress; P:catabolic process; P:cellular process; F:binding
High dosage treatment of ELICE16INDURES		
TRINITY_DN1035_c0_g1 TRINITY_DN623_c0_g1 TRINITY_DN1_c1_g1 TRINITY_DN2970_c0_g1 TRINITY_DN3188_c1_g1	peroxidase 1-like peroxidase 2-like peroxidase 5-like peroxidase 50-like	P:response to stress; P:catabolic process; P:cellular process; P:response to chemical; F:catalytic activity; F:binding; C:extracellular region
TRINITY_DN11193_c0_g1	auxin-responsive protein SAUR36-like	P:transport; P:signal transduction; P:response to endogenous stimulus; P:growth; P:response to chemical; C:cytoplasm
TRINITY_DN11681_c0_g1	pathogenesis-related protein PR-4-like (PR4) Acting on ester bonds	P:response to stress; P:response to external stimulus; P:response to biotic stimulus; F:nuclease activity
TRINITY_DN12031_c0_g1 TRINITY_DN2721_c0_g1	alpha-amylase/trypsin inhibitor-like	P:response to stress
TRINITY_DN12087_c0_g1	putative lipid-transfer protein	P:response to stress; P:response to external stimulus; P:response to biotic stimulus; F:lipid binding
TRINITY_DN1303_c0_g1	protein SRC2 homolog	P:response to stress; C:membrane
TRINITY_DN1308_c0_g5	pathogenesis-related protein (PR5)	P:response to stress; P:response to biotic stimulus
TRINITY_DN1552_c0_g1	subtilisin-chymotrypsin inhibitor-2A-like	P:response to stress; P:protein metabolic process; F:enzyme regulator activity
TRINITY_DN1626_c0_g2	barwin-like (PR4)	P:response to stress; P:response to external stimulus; P:response to biotic stimulus
TRINITY_DN21_c0_g1	putative wall-associated receptor kinase-like 16 Transferring phosphorus-containing groups	P:signal transduction; P:protein modification process; F:nucleotide binding; F:kinase activity; F:carbohydrate binding; C:membrane
TRINITY_DN4772_c0_g1	MLO-like protein 1	P:response to stress; P:response to biotic stimulus; F:protein binding; C:membrane
TRINITY_DN875_c0_g2	predicted protein	P:response to stress; C:nucleus
TRINITY_DN9594_c0_g1	pathogenesis-related protein 1-like (PR1)	P:response to stress; P:response to biotic stimulus; C:extracellular region
Low dosage treatment of ELICE16INDURES		
TRINITY_DN7081_c0_g2	nematode resistance protein-like HSPRO1	P:response to stress; P:catabolic process; P:cellular process; F:binding
TRINITY_DN914_c0_g1	glucan endo-1,3-beta-glucosidase GIII-like (PR2)	P:carbohydrate metabolic process; P:response to stress; F:hydrolase activity; C:plasma membrane
TRINITY_DN9142_c0_g3	ethylene-responsive transcription factor 2-like	P:signal transduction; P:response to endogenous stimulus; P:response to chemical; F:DNA binding; F:DNA-binding transcription factor activity; C:nucleus
TRINITY_DN2054_c0_g1	Subtilisin-chymotrypsin inhibitor-2B	P:response to stress; P:protein metabolic process; F:enzyme regulator activity

**Table 3 plants-11-02969-t003:** KEGG pathways of mapped genes with differential expression in pairwise analysis.

Pathways	Nr of KEGGs	Nr of Sequences	Nr of DEGs
High Dosage Treatment of ELICE16INDURES vs. Control			
Amino sugar and nucleotide sugar metabolism	ko00520	85	1
Alanine, aspartate and glutamate metabolism	ko00250	41	1
Carbon fixation in photosynthetic organisms	ko00710	54	1
Citrate cycle (TCA cycle)	ko00020	44	1
Fatty acid biosynthesis	ko00061	26	1
Fatty acid degradation	ko00071	30	1
Glycolysis/Gluconeogenesis	ko00010	94	1
Other glycan degradation	ko00511	33	2
Purine metabolism	ko00230	450	1
Pyruvate metabolism	ko00620	76	1
Thiamine metabolism	ko00730	402	1
Ubiquinone and other terpenoid-quinone biosynthesis	ko00130	59	1
Low dosage treatment of ELICE16INDURES vs. control			
Cutin, suberine and wax biosynthesis	ko00073	25	1
Drug metabolism-cytochrome P450	ko00982	53	1
Drug metabolism-other enzymes	ko00983	105	1
Glutathione metabolism	ko00480	58	1
Metabolism of xenobiotics by cytochrome P450	ko00980	48	1
Other glycan degradation	ko00511	33	1
Shared			
Phenylpropanoid biosynthesis	ko00940	73	9
Alanine, aspartate and glutamate metabolism	ko00250	41	1
Cyanoamino acid metabolism	ko00460	35	3
Fructose and mannose metabolism	ko00051	54	1
Starch and sucrose metabolism	ko00500	142	5

## Data Availability

All of the data supporting reported results can be found at Educomat Ltd. Keszthely, Hungary. NGS datasets are deposited in the NCBI sequence read archive database and Mendeley data.

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
