# Peer review of "Liposomal Formulation of Botanical Extracts may Enhance Yield Triggering PR Genes and Phenylpropanoid Pathway in Barley (Hordeum vulgare)"

_plants, 2022, doi:10.3390/plants11212969_

Round 1

Reviewer 1 Report

Overall it is a good paper, but I found a couple of things need to be addressed:

  1. Legend in the X axis in figure 2 (4l*ha-1). What is it?
  2. In the discussion section, The question, Why did two extremal treatments (20 and 240 g ha-1 ) show the most substantial effect? It is not addressed. Do the authors have a hypothesis?
  3. The authors suggested infection tests in further experiments could detect a connection with priming. Unfortunately, In field studies, the use of a biocide can not be overruled. 

Author Response

                                                                                           Debrecen, 23/10/2022

The Scientific Editor

Plants

Ref: Submission Manuscript ID: plants-1976972

Following your correspondence regarding the revision of the manuscript based on the recommendations of the reviewer, we thank you and appreciate the reviewers’ valuable input on improving the revised version of the manuscript. All the suggestions have been accepted, and the recommended additional information has been incorporated into the manuscript accordingly.

Article Title: "Liposomal formulation of botanical extracts may enhance yield
triggering PR genes and phenylpropanoid-pathway in barley (Hordeum vulgare)"

Remarks of the Editor:

Open Review 1:

  1. The field experiment was conducted only once, and the results were ambiguous. The authors do not explain why only the very low dose (20 g/ha) and the highest dose (240 g/ha) showed effectiveness in increasing the yield. The data from the field experiment should have been confirmed with a similar experiment in the phytotron. Plus, the dose should be increased because the highest dose (240 g/ha) showed the greatest effect.

Answer: Based on the literature we explained the effectiveness of low and high doses. According to this low dosage (20 g ha-1) may activate the elicitors of defensive genes (e.g. ethylene-responsive transcription factors and lead to primed plants) however high dosage (240 g ha-1) directly induces PR genes these roles take part in altered mechanisms of induced resistance. This explanation is incorporated into the ms. discussion and it is supported by similar results from the scientific literature. This explanation was summarized also in Figure 9.  Phytotron experiments are described in the ms. in which we found it expedient to use only those concentrations that showed a significant phenotypic difference in the field. Since the maximum dose of 60 g/ha is stated in the license document of the investigated agent, we received permission from the authorities for a maximum dose of 240 g/ha, therefore we could not increase the concentration more.

  1. In the last paragraph of the Introduction section, the goal and objectives of the work should be written more clearly, instead of stating the obtained results.

Answer: We wrote the objectives of this study much more clearly in the last paragraph of the Introduction section.

  1. 2C shows the different notations (% and kg/plots).

Answer: Fig 2 C shows the determination of mean values of crop measured in four parallel parcels in kg plots-1. To better understand, the yield enhancement is represented in percentage increase (%) in the diagram.  We incorporated this in the caption of the figure to avoid misunderstanding.

  1. Section 4.1, line 420 lists eleven botanical extracts, whereas Table 5 lists only 10.

Answer: We are sorry, one of the ingredients (Urtica dioica) was missing in Table 5, we replaced it in the table. However, the manufacturer requested the deletion of the ratios of extracts. Table 5 was moved into the supplementary material, as Table S6.

  1. The flow of material in whole section 4.2 needs to be streamlined. The authors should describe in detail the experimental design for the field experiment as well as the experiment in the phytotron.

Answer: We clarified the field and phytotron experimental procedures. To better understand, the field and phytotron experimental design was detailed and summarized in Figure 10.

  1. How necessary is it to give the abbreviation SC-CO2 in the title of the paper?

Answer: The abbreviation SC-CO2 was deleted from the title. However, we supplemented the title with ‘PR-genes’ which is an important finding of the ms.

The altered title is "Liposomal formulation of botanical extracts may enhance yield
triggering PR genes and phenylpropanoid-pathway in barley (Hordeum vulgare)"

Open Review 2:

  1. Abstract: This part should contain the main purpose of your study, which, in your case, is not clear. It is important that Abstract follows the structure of the manuscript. You can remove the first general sentence and replace it with a short phrase describing your aim. Please modify accordingly and check the whole abstract for the recommended number of words in the recommendations for authors.

Answer: thank you for your comment, we have rewritten the abstract to make it more comprehensible and introduce the aim of the study. We have taken the word limits.

  1. Introduction: You offer a well-documented background of your study but you need to compare the purposes of your study with similar ones and clarify what you bring in novelty. It is very important to state what exactly you bring in novelty in order to express your originality. The purpose of the study needs to be found in the last paragraph. Please add. In the present form, as neither Abstract, nor Introduction describe it, the manuscript lacks one of its most important parts. Please add further information and justifications and modify accordingly.

Answer: The Introduction section was supplemented with additional paragraphs discussing what is a novelty and reported firstly in the field, lines 81-95.

Additionally, based on the priming concept of low and high dosages discussed in the discussion and conclusion section we introduced the background information in this section, lines 96-110. We added the purpose of the study written in the last paragraph.

  1. Discussions: As you also compare your results with similar ones obtained in literature, you should state once again what you study brings in novelty, this time in terms of results.

Answer: The discussion section was completed in detail and explain better with the novelty of this study. Line 437-481. To better understand we incorporated figure 9, which summarizes the concept of low and high doses as the originality of the investigated agent.

  1. Conclusions: Please offer this section and include perspectives of your study. Your Results and Discussion sections are quite large and a Conclusions section is needed in order to extract the most important findings of your study.

Answer: The conclusion section was incorporated in the ms. summarizing the novel results and findings of the discussed experiments on Elice16Indures plant conditioner.

  1. References do not follow the recommendations of MDPI journals, please check Guidelines for Authors and modify accordingly.

Answer: References were modified according to the journal requirements.

We are grateful to the reviewer for helping us to shape the manuscript into its current form.

Thank you

Sincerely,

Géza Hegedűs, Barbara Kutasy, Kincső Decsi and Eszter Virág.

Reviewer 2 Report

The article considers the efficiency of barley leaves treatment with nanosized liposomes containing plant extracts of 11 species of medicinal plants in the doses from 10 to 240 g/ha. In field tests barley yield increase is shown when using doses of the preparation of 20 and 240 g/ha. A phytotron experiment showed up-regulation of the expression of a number of genes (including PR and TFs genes) in response to liposomal treatment.

The article contains new information, written in good language, documented with experimental data, graphs and diagrams.

Comments on the article.

1. The field experiment was conducted only once, and the results were ambiguous. The authors do not explain why only the very low dose (20 g/ha) and the highest dose (240 g/ha) showed effectiveness in increasing the yield. The data from the field experiment should have been confirmed with a similar experiment in the phytotron. Plus, the dose should be increased because the highest dose (240 g/ha)  showed the greatest effect.

2. In the last paragraph of the Introduction section, the goal and objectives of the work should be written more clearly, instead of stating the obtained results.

3. Fig. 2C shows the different notations (% and kg/plots).

4. Section 4.1, line 420 lists eleven botanical extracts, whereas Table 5 lists only 10.

5. The flow of material in whole section 4.2 needs to be streamlined. The authors should describe in detail the experimental design for the field experiment as well as the experiment in the phytotron. 

6. How necessary is it to give the abbreviation SC-CO2 in the title of the paper?

Author Response

(The authors gave the same response as above.)

Reviewer 3 Report

Dear Authors,

The present study evaluates how a liposomal formulation of SC-CO2 extracts may enhance yield triggering phenylpropanoid-pathway in Hordeum vulgare. The research subject is interesting, well documented and brings scientific important data in the field, as it deals with a subject that is currently of great interest. Some changes of the manuscript should nevertheless be performed in order to improve its quality. Following specific changes should thus be performed:

 Major changes

Abstract: This part should contain the main purpose of your study, which, in your case, is not clear. It is important that Abstract follows the structure of the manuscript. You can remove the first general sentence and replace it with a short phrase describing your aim. Please modify accordingly and check the whole abstract for the recommended number of words in the recommendations for authors.

Introduction: You offer a well-documented background of your study but you need to compare the purposes of your study with similar ones and clarify what you bring in novelty. It is very important to state what exactly you bring in novelty in order to express your originality. The purpose of the study needs to be found in the last paragraph. Please add. In the present form, as neither Abstract, nor Introduction describe it, the manuscript lacks one of its most important parts. Please add further information and justifications and modify accordingly.

Discussions: As you also compare your results with similar ones obtained in literature, you should state once again what you study brings in novelty, this time in terms of results.

Conclusions: Please offer this section and include perspectives of your study. Your Results and Discussion sections are quite large and a Conclusions section is needed in order to extract the most important findings of your study.

References do not follow the recommendations of MDPI journals, please check Guidelines for Authors and modify accordingly.

            All these suggested changes should be performed in order to bring further improvements to the manuscript. 

Author Response

(The authors gave the same response as above.)

Round 2

Reviewer 3 Report

Dear Authors,

The present study evaluates how a liposomal formulation of SC-CO2 extracts may enhance yield triggering phenylpropanoid-pathway in Hordeum vulgare. The authors performed most of the suggested changes in the first round of review. Following specific changes should still be performed:

 Minor changes

I find that novelty should be clearer. It is very difficult to follow changes, as they are not tracked. I cannot see where specific changes were performed. I think that both in Introduction and Discussion novelty is not clearly mentioned.

All these suggested changes should be performed in order to bring further improvements to the manuscript.

Author Response

Ref: Corrections of Manuscript ID: plants-1976972

Dear Editor,

We rephrased the Introduction of the manuscript, to avoid similarity with earlier works. The Materials and methods section still contains some similarities, but these are mainly methodological descriptions which we have also applied elsewhere, we thought it shouldn't be a problem for it to remain in this form.

The requested changes have been completed according to the reviewer's suggestions. Based on this we clarified the novelty of the work, involving an enumerated aim description in the Introduction section and the main findings are in the form of a list-type answer to it in the Discussion section.

To follow better the changes, we also attach the manuscript in form with the track of changes to the last version.

We hope that the manuscript is already in an acceptable form for publication.

Sincerely,

Kincső Decsi

***

Comments and Suggestions for Authors

Dear Authors,

The present study evaluates how a liposomal formulation of SC-CO2 extracts may enhance yield triggering phenylpropanoid-pathway in Hordeum vulgare. The authors performed most of the suggested changes in the first round of review. Following specific changes should still be performed:

 Minor changes

I find that novelty should be clearer. It is very difficult to follow changes, as they are not tracked. I cannot see where specific changes were performed. I think that both in Introduction and Discussion novelty is not clearly mentioned.

All these suggested changes should be performed in order to bring further improvements to the manuscript.

Answer: We clarified the novelty of the work, involving an enumerated aim description in the Introduction, and we describe the main findings as a list-type summary that also answers the aims in the Discussion. Please find:

Introduction:

In this study, we report on this agent as a potential new defense-priming material. Its technological development aimed to reach a 150-200 nm size multilamellar liposome formulation of eleven CO2-extracted medicinal plants as a novelty in this field.” lines 79-81.

In the present study, our goals were to i) develop 100-150 nm size MLV liposomes as a new formulation of 11 supercritical botanical extracts and their use as a biostimulant (named ELICE16INDURES); (ii) investigate the plant conditioning effect of this liposomal agent on the field population of autumn barley; (iii) for practical use, we aimed to determine dose-response by testing six concentrations and measuring yield parameters and photosynthetic activity; (iv) to understand underlying resistance mechanisms, genome-wide transcriptional profiling was in focus using the same barley test crops under controlled plant growth conditions; (v) in the phytotron experiment doses showing higher conditioning effects in the field were aimed to test and compare with each other to gain information on the comprehensive mechanism of action of this agent. Results obtained as these goals are considered new results for sustainable plant protection involving the field of new-generation biostimulant development.” lines 149-158.

Discussion:

To summarize our results and findings, (i) a 100-150 nm size MLV liposome-formulated botanical extracts-containing biostimulant, ELICE16INDURES, was developed and used in field practice; (ii) The plant conditioning effect of this agent was proved in four repetitions block system of autumn barley test crop; (iii) For practical use, dose-response of plants was determined showing two effective (extremal) dosages, 20 and 240 g ha-1, where the yield and the photosynthetic activity increased by 39.9% and 27.5%; (iv) We proved that these concentrations stimulate plant defense responses increasing the expressions of PR and phytohormone-signalling pathway genes and transcription factors; (v) Comparing the expressed genes and transcriptomic events between the two extremal dosages, we found that this biostimulant possess priming effect on the investigated test crop and may enhance ISR and SAR mechanisms depending on low and high concentrations.” lines 494-503.